# Neighborhood Socio-Economic Status Influences Motor Performance and Inhibitory Control in Kindergarten Children—Findings from the Cross-Sectional Kitafit Study

**DOI:** 10.3390/children10081332

**Published:** 2023-08-01

**Authors:** Nadja Schott, Andi Mündörfer, Benjamin Holfelder

**Affiliations:** 1Department of Psychology and Human Movement Sciences, Institute for Sport and Movement Science, University of Stuttgart, 70569 Stuttgart, Germany; benjamin.holfelder@inspo.uni-stuttgart.de; 2Amt für Sport und Bewegung, Bewegungsförderung und Sportentwicklung, 70161 Stuttgart, Germany; andi.muendoerfer@stuttgart.de

**Keywords:** probable Developmental Coordination Disorder, physical activity, motor skills, executive function, SES, kindergarten, sex

## Abstract

Numerous studies have examined the role of socio-economic status on physical activity, obesity, and cognitive performance in youth or older adults, but few studies have examined the role of neighborhood socio-economic status (NSES) on motor or cognitive performance in kindergarten children. This study aimed to examine whether lower NSES (measured by the social data atlas) was associated with lower motor and inhibitory control performance in kindergarten children. One hundred twenty-nine preschoolers were recruited from eight kindergartens in low and high NSES areas in Stuttgart, one of Germany’s largest metropolitan areas. Motor functioning (Movement Assessment Battery for Children, MABC-2; Manual Dexterity, Aiming and Catching, and Balance) and inhibitory control (Flanker Task, Go/NoGo Task) were assessed in a sample of 3- to 6-year-old children within a cross-sectional study. Children from a low NSES background showed the expected difficulties in inhibitory control and motor performance, as indicated by poorer performance than children from a high NSES background. Sex-specific analysis revealed girls from low NSES areas to have the lowest fine motor control; children with low NSES reach a Developmental Coordination Disorder at-risk status of 13% (boys and girls), in contrast to children with high SES (boys 9.1%, girls 0.0%). Motor performance and inhibitory control correlated positively with regard to the group from a low NSES background. Researchers and practitioners are advised to develop a more nuanced picture of motor and academic achievement in heterogeneous neighborhoods when designing early intervention programs, particularly with regard to sex differences, with the most significant disadvantage to girls with lower NSES.

## 1. Introduction

Socio-economic disadvantage (e.g., poverty, lack of prestige, place of residence) in early childhood, as experienced by 2.88 million German children in 2021, has profound and lasting effects on motor, cognitive, and mental health that can extend into adulthood [1,2,3,4,5].

At the individual level, socio-economic status (SES) is characterized as a multidimensional concept that is typically assessed using metrics such as occupation, educational attainment, and household income, each of which has different effects at different points in the life course [6]. For example, education has been shown to positively impact health by increasing knowledge of health-promoting behaviors and technologies while improving decision-making and problem-solving skills. Income and wealth, in turn, provide access to health-promoting resources (e.g., good quality housing in a safe environment) and quality health care [7].

In addition to the Body-Mass Index as an indicator of obesity [8], developmental health [9], physical activity [10,11], lower fitness [12,13,14], and neurodevelopmental disorders [15] have also been associated with lower SES. For example, children of mothers with high levels of education or from high-income households show relatively healthy lifestyles. In contrast, children of mothers with low education or from low-income households tend to show higher screen time and physical inactivity [11]. Socio-economic differences are also evident in muscular and cardiorespiratory fitness, particularly in girls, even before their decline in physical activity, typically seen around puberty [13]. A recent review further reported that low SES was associated with impaired motor development (Developmental Coordination Disorder; DCD). DCD can be defined as “[…] a neurodevelopmental disorder that affects children’s ability to execute coordinated motor actions, resulting in slow, clumsy, or inaccurate motor performances and learning difficulties (of new motor tasks or to adapt previously learned gestures to a modified or additional constraint)” ([16], p. 3).

Nevertheless, another study reported that higher SES increased the risk of motor impairment [17]. In cross-sectional studies, cognitive tests of executive function (EF), memory, and academic performance show strong to moderate linear associations with SES between individuals with low and high income [18,19,20]. Recently, longitudinal studies examining SES and cognitive development in childhood have proliferated. These studies show that children with low SES have lower baseline levels of EF [21,22]. Moreover, additional cross-sectional and longitudinal data suggest that SES differences are found in the brain structures that support EF. For example, lower SES has been associated with lower hippocampal and cortical volumes, including the prefrontal cortex [23,24,25]. While SES is related to academic performance and physical fitness, few longitudinal studies have examined the extent to which the relationship between physical fitness and academic performance varies by SES. London and Castrechini [26] published one of the few studies examining the relationship between physical fitness and academic achievement longitudinally, controlling for SES but not considering the interaction between SES and fitness. However, Clennin and colleagues [27] only recently showed better academic performance among fifth-grade students with higher SES than eighth-grade students with lower SES.

Some studies suggest that household and neighborhood SES (NSES; physical and social characteristics) should be considered separately. Physical characteristics refer to neighborhood attributes such as degree of urbanization (e.g., density), public and open spaces (e.g., walkability, transportation, cleanliness), available resources and amenities, green space, environmental noise (e.g., health care, safe streets, clean parks), and neurotoxic pollution. Social characteristics, for example, refer to factors such as deprivation, disorder (e.g., exposure to local violence), social cohesion, and ethnic composition [28]. Previous research has found strong associations between neighborhood characteristics, particularly disadvantage, and youth and older adults’ physical, behavioral, and mental health. For example, individuals in disadvantaged neighborhoods (i.e., lower social status and poor physical conditions) were, on average, at higher risk for obesity [29], poor peer relationships, lower cognitive development [5], and more mental and motor problems [30,31,32,33]. In contrast, adolescents who lived in “good” neighborhoods were more likely to play outside, watch less television, and engage in more activities that promote socialization and physical activity [34].

Although some aspects of the neighborhood have been associated with health, motor, and cognitive performance in older children [5,32,33], adults/and older adults, the results might differ from younger individuals [28]. As far as we know, no study to date has examined the relationship between neighborhood SES and motor and cognitive performance in kindergarten children. Because better motor and cognitive performance in early childhood is associated with academic performance [35], it is particularly important to understand the factors that influence this development in younger boys and girls.

Therefore, the main objective of this study was to determine whether neighborhood SES is associated with motor performance and inhibitory control in 3- to 6-year-old boys and girls. We hypothesize that differences in motor and inhibitory control performance would exist between kindergarten children with low and high NSES backgrounds. In addition, we assume that the association between motor and inhibitory control performance would be more robust for children with low NSES characteristics.

## 2. Materials and Methods

### 2.1. Participants

We conducted a cross-sectional study in Stuttgart, Germany, one of Germany’s largest metropolitan areas, by selecting eight kindergartens distributed throughout two types of areas in Stuttgart. The “Sozialdatenatlas” (social data atlas) profiles 152 districts by 31 indicators in eight types of areas („Gebietstypen“) with varying NSES. The data is updated approximately every three to four years and serves as a foundation for decisions between local policy and administration. We selected areas with low or high NSES characteristics as a stratification criterion. Only kindergartens in regions with high NSES (Type 1 and 2; few children, many older people, low proportion of migrants, plenty of housing, many playgrounds, good facilities in kindergartens and schools; approx. 40 places in 2 to 3 groups per kindergarten; response rate 40%) and low NSES (Type 6 and 7; families with many children, a high proportion of migrants, poverty, fewer playgrounds, low educational opportunities, poorer equipment in kindergartens and schools; approx. 20 to 55 places in 2 to 3 groups per kindergarten; response rate 38%) were therefore asked to participate in the study.

A priori power analysis using G*Power [36] to detect a medium effect size (f = 0.25) with a power (1-β) of 0.80 and an α = 05 resulted in an overall estimated sample size of 128 children. We randomly selected 129 children (65 girls, 64 boys) aged 3 to 6 years. Any child with a diagnosed severe sensory, physical, or intellectual disability known to the teachers was excluded from the study. Four kindergartens were located in high NSES areas and four in low NSES areas. The kindergartens are served by different organization types (state, church, sports club, unorganized). The children were asked for their consent and willingness to participate in the study. The participant’s legal guardian/next of kin provided written informed consent to participate in this study. The participants or the legal guardians of the children did not receive any incentive for participating in the study. The Ethical Review Board has approved this study. All procedures were in accordance with the Declaration of Helsinki, with ethical standards, legal requirements, and international norms.

### 2.2. Instruments

#### 2.2.1. Motor Assessment

Motor performance was evaluated using the Movement Assessment Battery for Children 2nd edition (MABC-2; [37]), encompassing three components measuring different aspects of motor functioning: Manual Dexterity (posting coins, threading beads, and bicycle trail), Aiming and Catching (catching a beanbag and throwing a beanbag at a target), and Balance (one-leg balance, walking heels raised and jumping on mats). For each component and the total test (sum of three component scores), the sums of points are transformed into age-adjusted standard scores and percentiles (according to German norms), with higher scores indicating better performance. A score up to the 5th percentile indicates motor difficulty (probable DCD), a score between the 5th and 16th percentile indicates suspected motor difficulty and a score above the 16th percentile suggests typical motor performance.

This tool has been designed to identify motor impairments in children aged 3 to 16 years and is one of the most commonly used and recommended tests for assessing motor impairment [38]. In this study, Cronbach’s alpha value for all standardized items was 0.76.

#### 2.2.2. Inhibitory Control Assessment

Stimuli for the two tests were presented on a 15-inch monitor using E-Prime 2.0 software (Psychology Software Tools, Inc., Pittsburgh, PA, USA), with participants seated approximately 50 cm from the screen. Participants were instructed to respond as quickly as possible for both tasks while minimizing their mistakes. Responses were recorded using a standard QWERTY keyboard.

*Go/NoGo Task.* Inhibitory control (simple response) was assessed by performance during a modified Go/NoGo Task. As a standard test [39], the Go/NoGo Task is simple to administer and is sensitive as a simple inhibition task without interference. During the task, participants were presented with a target stimulus (dog) in different colors (two blocks, each with 36 trials), the Go-stimuli in three different colors (75% of all trials), requiring the participants to press the blue key (space bar with a round blue sticker), and the NoGo-stimulus in one color (25% of all trails), requiring the participants to inhibit the pressing impulse and to press none of the keys. All analyses excluded individual trials with reaction times (RT) outside the 200–1650 ms post-stimulus onset window and incorrect trials from the RT analysis [40].

*Flanker Task.* The experimental paradigm for investigating complex response inhibitory control consisted of a modified Flanker Task [41]. In our task version, participants were presented with a horizontal array of five fish on each trial, and they had to indicate whether the fish in the center of the display was swimming with the other fish in the same or opposite direction. In the congruent version of the task (all fish “swim“ in the same direction), participants had to press the yellow button (“C” with a round yellow sticker). In the incongruent condition (surrounding fish swimming in the opposite direction), participants were required to press the green button (“M” with a round green sticker). All stimuli were presented for 1000 ms against a white background. A randomized inter-stimulus interval of 1500 to 2000 ms was used. There were, in total, 80 trials, including 16 practice trials and two experimental blocks with 32 trials each. A brief break and encouragement were given between each block. Both the number of trials within each condition and the frequency of target direction were randomized with equal probability. For all analyses, individual trials with RTs outside the 200–1650 ms post-stimulus onset window and incorrect trials were excluded from the RT analysis [40]. Due to technical reasons, we can only report the results for the incongruent trials here.

### 2.3. Procedure

The children of each kindergarten were assessed from 9 to 12 p.m. on different days by a Ph.D. student and two master’s students. The complete assessment lasted approximately 40 min (motor and inhibitory control assessments 20 min each). The children had a short break between the motor and inhibitory control assessments. The motor assessment took place in a room for physical education, and the inhibitory control assessment took place in a separate room to realize a quiet and non-distracting environment.

### 2.4. Data Analysis

Analyses were performed using SPSS (Version 27; SPSS, Inc., Chicago, IL, USA). Descriptive analyses (mean, standard deviation, range, frequencies) were used to describe the demographic data and measure performance (MABC-2, Flanker Task, Go/NoGo Task).

Intergroup differences between the two groups with low and high NSES backgrounds were examined by MANOVA, with the subdimensions of the MABC-2 being the dependent variables. For the two inhibitory control tasks, we excluded all responses that were 2.5 standard deviations above or below the children’s individual mean from the analyses. We calculated accuracy as the percentage of correct responses and response times using the mean response times for the correct responses. A series of covariance (ANCOVAs) analyses were conducted to determine the between-subject effect of NSES background and sex, controlling for age. In addition to *p* values, effect sizes as measured by partial Eta Squared (η_p_^2^) values were used for data interpretation. According to Cohen [42], only η_p_^2^ of ≥0.14 obtained from MANCOVA analyses are considered sufficiently large for practical meaning. Finally, linear hierarchical regression models were conducted to determine the relevant predictors (age, sex, NSES, and motor performance (MABC-2 composites and total score) of inhibitory control.

## 3. Results

### 3.1. Participants

The demographic characteristics of the sample are presented in Table 1. The mean age of children was 60.7 months (SD = 7.63 months). In particular, among children from low NSES families, only one in two actively participated in sports activities in the club (60% of the boys, 40% of the girls). In contrast, over 75 percent of children from high-NSES areas are members of sports clubs (88% of the boys, 78% of the girls).

### 3.2. Motor Performance

Table 2 presents the means and standard deviations of the children’s scores on each test item, averaged across NSES and sex. A MANOVA was undertaken to investigate the effects of NSES characteristics and sex on test performance. This analysis revealed a statistically significant effect of NSES, Wilks’ Lambda = 0.84, *F*(4, 119) = 5.63, *p* < 0.001, and NSES by sex, Wilks’ Lambda = 0.90, *F*(4, 119) = 3.22, *p* = 0.015 with an overall moderate effect size of 0.159 and 0.098, respectively. Follow-up, univariate analysis of the main effect of NSES showed that this was held only for manual dexterity, with children from a background with high NSES performing faster and more accurately than children from low NSES, *F*(1, 122) = 8.12, *p* = 0.005, η_p_^2^ = 0.062). However, there was also a tendency to show better results for the balance tests for children from high NSES vs. low NSES areas, *F*(1, 122) = 2.74, *p* = 0.100, η_p_^2^ = 0.022).

Further analysis of the interaction of NSES and sex yielded statistically significant effects on manual dexterity, *F*(1, 122) = 13.1, *p* = 0.001, η_p_^2^ = 0.097), and the total score (*F*(1, 122) = 10.2, *p* = 0.002, η_p_^2^ = 0.077). However, it approached significance for balance (*F*(1, 122) = 3.51, *p* = 0.064, η_p_^2^ = 0.028). While boys with low NSES performed better than boys from high NSES, especially in aiming and catching, girls from low NSES performed worse in manual dexterity and balance than girls from high NSES. Furthermore, more children with probable DCD were from low NSES areas (13.1 vs. 4.6%; *CHI*^2^(1) = 2.85, *p* = 0.084). This was especially true for girls (*CHI*^2^(1) = 4.41, *p* = 0.036).

Additional analysis addressing the influence of the membership in a sports club revealed better descriptive performances for those children who are active members; however, we observed only for balance a significant difference (percentiles: 57.2 vs. 73.3), *t*(86) = −2.52, *p* = 0.014, d = −0.54.

### 3.3. Inhibitory Control

Table 3 presents the accuracy and RT (means ± SD) of the children’s inhibitory control performance on each test item, averaged across NSES and sex.

NSES and Sex were entered as between-subject factors for the analyses of variance on accuracy and RT, controlled for age. The main effect of NSES was significant for both accuracy (Flanker Task: *F*(1, 104) = 7.71, *p* = 0.007, η_p_^2^ = 0.069; Go/NoGo Task: *F*(1, 109) = 6.54, *p* = 0.012, η_p_^2^ = 0.057) and RT (Flanker Task: *F*(1, 104) = 4.31, *p* = 0.040, η_p_^2^ = 0.040). Overall, children from high NSES backgrounds were more accurate but slower than children from low NSES backgrounds. There were no main effects of sex on RT or accuracy, nor was there an interaction between NSES background and sex. In addition, children of increasing age produce higher accuracy rates and lower RTs on both tasks.

### 3.4. Associations between Motor and Inhibitory Control Performance

The correlations between all scores in the complete dataset are illustrated in Table 4. Note that correction for multiple comparisons has not been applied. An appropriate Bonferroni threshold for 16 comparisons would be *p* < 0.003. While the correlations between manual dexterity, total score, and the Flanker Task for the children from a high NSES background were significant, the correlations between balance, total score, and both inhibitory control tasks were significant for the children from a low NSES background. The analysis revealed that manual dexterity was positively associated with Flanker Task accuracy and negatively associated with Flanker Task RT, suggesting that children with better motor performance and a high NSES background exhibited better Flanker Task performances. In children from low NSES backgrounds, a better balance was associated with better accuracy on the Flanker Task and the Go/NoGo Task.

Additional analyses show that children with very low motor performance (probable DCD) achieve significantly lower accuracy scores (Go/NoGo Task: 80.7 ± 11.9 vs. 93.0 ± 7.53; *t*(109) = −4.02, *p* < 0.001, d = −1.57; Flanker Task: 52.8 ± 31.0 vs. 74.5 ± 22.3; *t*(104) = −2.09, *p* = 0.040, d = −0.96) in the inhibitory control tasks.

Multiple linear regression models on inhibitory control were conducted with age, sex, NSES, and motor performance (MABC-2 composites) as predictor variables (Table 5). The results showed that age significantly predicted inhibitory control (accuracy and RT); NSES, balance, and manual dexterity were significant predictors of accuracy, with the effects being stronger for girls than for boys.

## 4. Discussion

In recent decades, the number of research studies examining the relationship between movement and cognition has increased [43]. In addition, most studies investigating motor and cognitive skills targeted school-aged children (>6 years) and academic performance in the classroom [44]. To our knowledge, this is the first study to examine the relationship between motor skills, inhibitory control, and neighborhood characteristics in kindergarten-age children. We hypothesized that, first, differences in motor and inhibitory control performance would exist between kindergarten children with low and high NSES backgrounds and, second, the association between motor and inhibitory control performance would be more robust for children with low NSES characteristics. The results suggest that neighborhood-level SES is significantly related to developmental outcomes in preschool-aged children. NSES was most strongly associated with manual dexterity, exercise (club membership), and response accuracy in inhibitory control and was weakest in the domain of ball skills.

Our results showed several associations between neighborhood characteristics and kindergarten children’s motor and inhibitory control outcomes. First, the better performances of girls in manual dexterity and balance and locomotor tasks are consistent with other studies. The same is valid for boys’ higher performance in ball skills [45,46]. However, the focus of the findings on motor performance was to investigate possible sex differences in children with heterogeneous NSES backgrounds. As diverse as the neighborhoods are in terms of their indicators in the Social Data Atlas, the results of the NSES perspective compared to the sex perspective was an additional focus of the examination. The significant difference in the MABC-2 manual dexterity, balance skills, and the total score can be explained by higher levels of family support and club sports activities. In their meta-analysis, Barnett et al. [47] found seven studies in which the child’s socioeconomic background was positively related to locomotion, stability, and dexterity. Hardy et al. [48] p. 506 conclude that “[…] biology does not fully explain sex differences […]” but that sex differences are more likely associated with children’s socialization. In our study, girls from high NSES areas showed better manual dexterity than boys from high NSES areas.

Interestingly, the results are reversed for children from low NSES backgrounds, with girls performing worse on the MABC-2 manual dexterity subtest. Fine motor performance has been previously related to environmental variables [49]. Therefore, educators in kindergartens of high NSES backgrounds may be more aware of the importance of exposing children to extracurricular activities and games focusing on fine motor skill development, thereby improving their school readiness. For example, socioeconomic status has already been shown to predict school-age fine motor performance [50].

In addition, we find a higher proportion of children with probable DCD in areas with low NSES (13.1%) compared with areas with high NSES (4.6%), consistent with the findings of Lingam et al. [51]. This difference is mainly caused by the high NSES group of girls and their superior hand dexterity, although no girl with probable DCD was identified in this group. An enriching environment that provides safe and supportive learning opportunities enhances children’s development through the availability of and access to resources. Since deficits in skill acquisition characterize DCD, it is hypothesized that the manifestation of motor difficulties in children with DCD will also depend on the opportunities provided by the environment—in this case, the kindergarten and its immediate environments [52].

Second, our study showed a relationship between low NSES and inhibitory control performance in a Flanker Task and a Go/NoGo Task. Our findings fit in with several previous studies on NSES, brain activity, and cognitive functioning in preschool and school-aged children and adolescents that have demonstrated (N)SES-related differences in cognitive functioning [53,54,55,56,57].

For example, Wei et al. [57] found that higher NSES disadvantage was cross-sectionally associated with lower inhibitory performance in children at age 4. Studies also suggest that children growing up in lower social status environments have less access to quality educational resources in terms of kindergarten and school funding and physical infrastructure (e.g., toys to promote fine motor skills; built environment [playgrounds; green space]), as well as increased exposure to environmental stressors such as noise, heat, and air pollution [54,58]. For example, several recent studies have found a positive association between neighborhood greening and children’s scores on cognitive tests in urban and suburban areas [58]. These associations are consistent with bioecological, biophysical, biocognitive, and biosocial theories [59] that cognitive stimulation in direct and indirect environments can enhance cognitive development, including EF.

Third, the result of our study on the relationship between fine motor skills and inhibitory control is consistent with existing studies [60,61,62]. The results from the regression analysis of manual dexterity and NSES background revealed that the lower motor performance of children from a low NSES background goes hand in hand with poorer inhibitory control performance. Regarding the relationship between motor and various aspects of cognitive performance, the literature found that coordinative exercises increase concentration and attention [63], fine motor skills predict academic performance [60,64], balance correlates with working memory [65], and ball skills correlate with cognitive performance [61,65]. Verdine et al. [66] consider fundamental movement skills as an essential indicator of school readiness, especially in mathematics and reading. Roebers et al. ([64], p. 294) point to the stable importance of EF over a two-year period, “[…] when fine motor skills, non-verbal intelligence, and executive functioning were integrated in one model.” In our study, not only do children from high NSES areas perform better in manual dexterity and accuracy on the Flanker Task, but based on univariate analysis, it is likely that the high SES children would perform better in manual dexterity than the other children at the same age.

This study has limitations; a larger sample would have provided more meaningful and generalizable results across various additional domains such as different age groups as well as neurodevelopmental disorders (e.g., probable DCD, Asperger, ADHD). Another limitation is the average age of the SES groups, which could have been more even. In addition, there is an age difference of six months between the children in the different SES groups. In this age range, significant development and performance shifts are expected in the younger children up to the same age. Another limitation relates more to testing at the elementary level. Although the examiner was careful to create a calm and motivating testing environment, the (very) young age of the participants may have led to distractions. Furthermore, it should be considered that the study was conducted in an urban setting with large differences in lifestyle, social support, and financial opportunities. However, the study aimed to collect data in a very heterogeneous environment, so it is impossible to get a simple picture of the situation.

## 5. Conclusions

The present study provides results in three areas: First, it confirms previous findings on the importance of manual dexterity as a strong predictor of academic achievement concerning school readiness [50], as well as recent studies on mathematics and reading achievement [66]. Second, it confirms the recent findings of Cameron et al. [60] and Roebers et al. [64], which showed the importance of fine motor and EF. Moreover, third, the present study adds to the knowledge about sex and NSES differences in inhibitory control and motor performance in kindergarten age. Summarizing the findings of this study, it is evident that the environment in high NSES areas from 3 to 6 years of age provides more profitable opportunities for personal development and that these opportunities make a significant difference in motor and inhibitory control performance. One aspect demonstrated in this study is membership in sports clubs, which is higher in high NSES areas. A possible consequence is that children with low NSES reach a DCD potential of 13% (boys and girls), in contrast to children with high SES (boys 9.1%, girls 0.0%).

Finally, the results show that considering sex alone masks diversity in motor and inhibitory control performance. The overall developmental differences in preschool children cannot be unmasked without considering sex as a moderator in a particular setting. Researchers and practitioners are advised to develop a more nuanced picture of motor and academic achievement in heterogeneous neighborhoods when designing early intervention programs, particularly with regard to sex differences with the most significant disadvantage to girls in lower NSES areas. Assuming that future studies confirm these findings, they could make an important contribution to urban planning regarding the development of children and adolescents. Although the association system in Germany is characterized by volunteers and relatively inexpensive membership fees, a lack of staff and financial resources lead to fewer “low-threshold sports club activities” accessible to people from areas with a low NSES. This means that more volunteering and financial support for these exercise programs would be desirable.

## Figures and Tables

**Table 1 children-10-01332-t001:** Group characteristics of children from lower and higher social-economic backgrounds (*n* = 129).

	High NSES (*n* = 68)	Low NSES (*n* = 61)	Statistical Analysis
Age (years)	4.82 ± 0.63	5.39 ± 0.53	*t*(127) = −5.48, *p* < 0.001, d = −0.97
Sex (*n*)	34 boys, 34 girls	30 boys, 31 girls	*CHI*^2^(1) = 0.09, *p* = 0.926
Twins (yes in %)	11.8	9.8	*CHI*^2^(1) = 0.12, *p* = 0.725
Exercise (Membership in a formal sports club, yes in %)	83.3	50	*CHI*^2^(1) = 11.2, *p* = 0.001

**Table 2 children-10-01332-t002:** Mean scores of the Movement Assessment Battery for Children (percentiles) by sex and social milieu, mean ± SD.

	High NSES		Low NSES		Statistical Analysis		
	Boys	Girls	Boys	Girls	NSES	S	NSES × S
Manual dexterity	47.2 ± 35.0	71.6 ± 25.1	51.3 ± 31.2	37.1 ± 27.0	**	-	**
Aiming and Catching	60.7 ± 26.7	60.5 ± 23.8	71.2 ± 25.9	56.7 ± 28.1	-	-	-
Balance	64.9 ± 26.9	79.0 ± 25.7	66.0 ± 25.9	61.8 ± 30.5	T	-	T
Total score	56.8 ± 33.0	76.4 ± 23.1	66.4 ± 28.6	53.4 ± 28.8	-	-	**
pDCD (%)	9.1	0	13.3	12.9	T	-	-

Note: T *p* < 0.10; ** *p* < 0.001; NSES—neighborhood social-economic status; pDCD—probable Developmental Coordination Disorder; S—sex.

**Table 3 children-10-01332-t003:** Mean scores for inhibitory control performances by sex and social milieu, mean ± SD.

	High NSES		Low NSES		Statistical Analysis		
	Boys	Girls	Boys	Girls	NSES	S	NSES × S
*Flanker Task*							
Accuracy (%)	71.7 ± 26.6	77.6 ± 19.8	69.7 ± 24.0	75.5 ± 20.1	**	-	-
RT (ms)	1602 ± 320	1593 ± 249	1361 ± 191	1425 ± 234	*	-	-
*Go/NoGo Task*							
Accuracy (%)	92.4 ± 7.30	94.3 ± 6.89	91.1 ± 9.14	91.0 ± 9.59	*	-	-
RT (ms)	767 ± 193	731 ± 125	652± 97.4	736 ± 182	-	-	-

Note: * *p* < 0.05; ** *p* < 0.001; NSES—neighborhood social-economic status; RT—reaction time; S—sex.

**Table 4 children-10-01332-t004:** Partial Correlations between all variables for children with low and high neighborhood social-economic status; controlled for age and sex. # indicates correlations meeting the *p* < 0.003.

		Flanker Task	Flanker Task	Go/NoGo Task	Go/NoGo Task
		Accuracy (%)	RT (ms)	Accuracy (%)	RT (ms)
Manual dexterity	High NSES	0.417 **^#^	−0.353 *	0.223	−0.241
	Low NSES	0.354 *	−0.115	0.332 *	−0.214
Aiming and Catching	High NSES	0.038	−0.254	0.056	−0.05
	Low NSES	0.346 *	−0.145	0.523 **^#^	−0.378 **
Balance	High NSES	0.290 *	−0.093	−0.094	−0.146
	Low NSES	0.620 **^#^	−0.493 **^#^	0.599 **^#^	−0.488 **^#^
Total score	High NSES	0.391 **	−0.347 *	0.113	−0.252
	Low NSES	0.594 **^#^	−0.262	0.583 **^#^	−0.481 **^#^

Note: * *p* < 0.05; ** *p* < 0.001; ACC—accuracy; NSES—neighborhood social-economic status; RT—reaction time.

**Table 5 children-10-01332-t005:** Linear regression models to determine the relevant predictors (age, sex, neighborhood social-economic status, and motor performance (MABC composites and total score)) of inhibitory control separated by sex.

Dependent Variable	Sex	Predictors	*β*	*t*	*p*	Δ*R*^2^	Adjusted *R*^2^
Flanker Task Accuracy (%)	Boys	Age	0.552	4.07	<0.001	0.149	0.263
		NSES	0.347	2.55	0.014	0.074	
		Manual dexterity	0.286	2.46	0.017	0.081	
	Girls	Age	0.443	4.48	<0.001	0.125	0.561
		Balance	0.39	3.7	<0.001	0.34	
		Manual dexterity	0.447	4.39	<0.001	123	
Flanker Task RT (ms)	Boys	NSES	0.437	3.57	<0.001	0.191	0.176
	Girls	Age	−0.559	−4.85	<0.001	0.32	0.349
		Balance	−0.237	−2.06	45	0.056	
Go/NoGo Task Accuracy (%)	Boys	Age	0.415	3.48	<0.001	0.172	0.158
	Girls	Age	0.394	3.41	0.001	0.093	0.377
		NSES	0.134	1	0.32	0.152	
		Balance	0.288	2.3	0.025	0.131	
		Manual dexterity	0.291	2.03	0.048	0.046	
Go/NoGo Task	Boys	Age	−0.41	−3.43	0.001	0.168	0.154
RT (ms)	Girls	-	-	-	-	-	-

Note: NSES—neighborhood social-economic status; RT—reaction.

## Data Availability

Data are available from the authors upon request.

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
