# Peer review of "Neighborhood Socio-Economic Status Influences Motor Performance and Inhibitory Control in Kindergarten Children—Findings from the Cross-Sectional Kitafit Study"

_children, 2023, doi:10.3390/children10081332_

Round 1

Reviewer 1 Report

Comments

This study aims to examine the cross-sectional association of neighborhood socio-economic status with motor performance and inhibitory control in kindergarten children.

The manuscript has several issues that need to be addressed.

1. The authors should clarify the design of the study in the title and abstract. In addition, it would be better to avoid using any term (for example, “predict” in line 296) that indicates the temporal relationship between the exposure and outcomes throughout the manuscript.

2. It should be noted that “inhibitory control” and “cognitive function” are not interchangeable terms. I would suggest to use “inhibitory control” as the one of the outcomes throughout the manuscript.

3. The introduction would benefit from some rewriting to make the logic flow more coherently and provide clear background for the specific research question of this study. For example, BMI and physical activity are not really the focus of this study.

In addition, I am confused about the statement in lines 103-105, because there is no physical activity or sedentary behavior data in this study.

4. Please provide a reference for line 97 and the exact age of the sample in this study.

5. Line 195-196, please provide the number of subjects excluded from this analysis.

6. Line 191, “M-ABC-2” should be “MABC-2”.

7. Please provide the definition DCD in the Method.

8. I didn’t see how Tables 4-5 are relevant to the purpose of this study. I also did see why motor function is treated as the independent variable in Table 5. In addition, the linear regression model should be sex-specific given the interaction of sex.

9. I am confused about the statements in Lines 287-291 given this study was aimed to examine the association of neighborhood SES with motor performance and inhibitory control.

10. Although the author mentioned the small sample size as one of the limitations, it is unclear how the small sample size will affect the current findings.

Reviewer 2 Report

Thank you for opportunity of reviewing this manuscript. 

In the introduction, it is necessary to reduce the already widely known results of SES and expand on the need to learn about SES in the neighborhood.

Please provide a specific research question or hypothesis.

The meaning of 'severe disability' is very subjective. Criteria for exclusion and inclusion of participants should be described in detail.

Information is needed on who made the measurements and who was qualified for measurements.

The discussion centered on the research results is well described, but it seems to lack content about the implications that the results can provide. Based on the research results, it is necessary to suggest implications, directions for developing programs and policies for children, and suggestions for those who raise or educate children.

Round 2

Reviewer 2 Report

Revisions was made according to my comments. 

Thank you for your effort. 

Author Response

Thank you very much for your feedback.